# Pharmacotherapy for Advanced Non-Small Cell Lung Cancer with Performance Status 2 without Druggable Gene Alterations: Could Immune Checkpoint Inhibitors Be a Game Changer?

**DOI:** 10.3390/cancers14194861

**Published:** 2022-10-05

**Authors:** Satoshi Ikeda, Tateaki Naito, Satoru Miura, Kentaro Ito, Naoki Furuya, Toshihiro Misumi, Takashi Ogura, Terufumi Kato

**Affiliations:** 1Department of Respiratory Medicine, Kanagawa Cardiovascular and Respiratory Center, 6-16-1, Tomioka-higashi, Kanazawa-ku, Yokohama 236-0051, Japan; 2Division of Thoracic Oncology, Shizuoka Cancer Center, 1007, Shimonagakubo, Nagaizumi-cho, Shizuoka 411-8777, Japan; 3Department of Internal Medicine, Niigata Cancer Center Hospital, 2-15-3, Kawagishi-cho, Chuo-Ku, Niigata 951-8566, Japan; 4Respiratory Center, Matsusaka Municipal Hospital, 1550, Tonomachi, Matsusaka 515-0073, Japan; 5Division of Respiratory Medicine, Department of Internal Medicine, St. Marianna University School of Medicine, 2-16-1, Sugao, Kawasaki 216-8511, Japan; 6Department of Data Science, National Cancer Center Hospital East, 6-5-1, Kashiwanoha, Kashiwa 277-8577, Japan; 7Department of Thoracic Oncology, Kanagawa Cancer Center, 2-3-2, Nakao, Asahi-ku, Yokohama 241-8515, Japan

**Keywords:** non-small cell lung cancer, performance status, cytotoxic chemotherapy, immune checkpoint inhibitor, cancer cachexia

## Abstract

**Simple Summary:**

Data on the efficacy and safety of pharmacotherapy for advanced non-small cell lung cancer (NSCLC) with poor performance status (PS) 2 are insufficient. Cytotoxic chemotherapy for patients with PS 2 is insufficiently effective, and there are concerns about toxicity. Immune checkpoint inhibitors are a promising treatment with the potential for less severe toxicity, but data are more limited than with cytotoxic chemotherapy. In this review article, we summarize the current evidence on pharmacotherapy for NSCLC patients with PS 2 and without druggable genetic alterations, and we discuss future perspectives and challenges.

**Abstract:**

Most pivotal clinical trials in advanced non-small cell lung cancer (NSCLC) have excluded patients with poor performance status (PS), and data on the efficacy and safety of pharmacotherapy have not been fully accumulated. For NSCLC patients with PS 2 and without druggable genetic alterations, monotherapy with cytotoxic agents or carboplatin-based combination therapy is usually administered based on the results of several randomized trials. However, the evidence of cytotoxic chemotherapy for patients with PS 2 is insufficient, with limited efficacy and toxicity concerns. Immune checkpoint inhibitors (ICIs) are a promising treatment for patients with PS 2 because of lower incidence of severe toxicity compared to cytotoxic chemotherapy. Meanwhile, several reports suggest that anti-PD-1 antibodies monotherapy is less effective for patients with PS 2, especially for those with PS 2 caused by disease burden. Although the combination therapy of nivolumab and ipilimumab is a promising treatment option, there is a divergence in efficacy data between clinical trials. The standard of care for advanced NSCLC with PS 2 has not been established, and future therapeutic strategies should take into account the heterogeneity of the PS 2 population.

## 1. Introduction

Non-small cell lung cancer (NSCLC) accounts for about 85% of all lung cancers, with about 40% of them at stage IIIB—IV. Of these, approximately 10–18% have an Eastern Cooperative Oncology Group (ECOG) performance status (PS) of 2 at the time of diagnosis [1,2]. Poor PS is an important poor prognostic factor in advanced NSCLC and increases the incidence of various adverse events and treatment-related deaths caused by pharmacotherapy [3]. The ECOG PS scale is simple and widely used in clinical practice as important information for treatment decision making. On the other hand, there are some challenges; (1) it is clinician based and therefore sometimes lacks objectivity [4], (2) the causes of poor PS are varied (e.g., disease burden of lung cancer itself, age-related changes, nutritional status, comorbidities, etc.), so the patient population with PS 2 is heterogenous.

Most pivotal clinical trials in advanced NSCLC have excluded patients with PS 2, and data on the efficacy and safety of pharmacotherapy have not been fully accumulated. However, for NSCLC patients with PS 2 harboring druggable genetic alterations, various targeted therapies may be relatively safe treatment options. Studies comparing targeted therapy with cytotoxic chemotherapy in NSCLC patients with EGFR mutations or ALK fusion genes have included 5–10% of patients with PS 2 and have shown efficacy comparable to PS 0-1 [5,6,7,8]. In addition, gefitinib for patients with EGFR mutations and alectinib for patients with ALK fusion gene have also been reported to be effective in interventional studies for the patients with PS 2 [9,10]. Data on PS 2 patients with rare druggable genetic alterations (e.g., EGFR uncommon mutation, ROS1 fusion gene, BRAF gene mutation, MET gene mutation, RET fusion gene) are limited due to small number of patients, but in light of the results with EGFR tyrosine kinase inhibitors and ALK inhibitors, we may recommend that patients be treated with targeted therapies.

Meanwhile, for NSCLC patients with PS 2 and without druggable genetic alterations, monotherapy with cytotoxic agents or carboplatin-based combination therapy is usually administered based on the results of several randomized trials, which are discussed in more detail in later sections. However, the evidence of cytotoxic chemotherapy for patients with PS 2 is insufficient, with limited efficacy and toxicity concerns. Immune checkpoint inhibitors (ICIs) are a promising treatment with the potential for less severe toxicity, but data are more limited than with cytotoxic chemotherapy, with many issues to be resolved. 

In this review article, we summarize the current evidence on pharmacotherapy for NSCLC patients with PS 2 and without druggable genetic alterations, and we discuss challenges and future perspectives with a particular focus on ICI. As a reminder, strictly speaking, most of the trials discussed in this review article do not require measurement of rare druggable genetic alterations (e.g., MET, RET, NTRK, etc.) other than the relatively frequent ones such as EGFR and ALK. However, all trials exclude patients who have been previously treated with targeted therapies.

## 2. Current Evidence on Pharmacotherapy for NSCLC with PS 2

### 2.1. Cytotoxic Chemotherapy

In a meta-analysis of studies comparing monotherapy with cytotoxic agents (docetaxel, paclitaxel, vinorelbine, gemcitabine, etc.) versus best supportive care, pharmacotherapy prolonged overall survival and increased 1-year survival rates in approximately 30% of patients with PS ≥ 2 [11]. 

In addition, several randomized trials comparing the efficacy and safety of platinum-based combination therapy with cytotoxic anticancer agent monotherapy have been reported [12,13,14,15,16,17,18] (Table 1). In a subgroup analysis of PS 2 in the phase III CALGB9730 trial comparing the combination therapy of carboplatin and paclitaxel with paclitaxel monotherapy, the combination therapy was superior to monotherapy in 1-year survival (18% vs. 10%, Hazard ratio [HR] 0.60, 95% confidence interval [CI]: 0.40–0.91) [12]. The study comparing the combination therapy of carboplatin and gemcitabine with gemcitabine monotherapy showed a trend toward longer overall survival (OS) (6.7 months vs. 4.8 months, *p* = 0.49) and progression free survival (PFS) (4.1 months vs. 3.0 months, *p* = 0.36) in the combination therapy group, although the results were not statistically significant [13]. Furthermore, in a phase III trial of carboplatin plus pemetrexed versus pemetrexed monotherapy in advanced NSCLC with PS 2, the combination therapy group showed significantly longer PFS (5.8 months vs. 2.8 months, HR 0.46, 95% CI: 0.35–0.63, *p* < 0.001) and significantly longer OS (9.3 months vs. 5.3 months, HR 0.62, 95% CI: 0.46–0.83, *p* = 0.001). Meanwhile, with regard to toxicity, 3.9% of treatment-related deaths were observed in addition to a higher frequency of anemia and neutropenia in the combination therapy group [18]. 

Based on these results, monotherapy with cytotoxic agents or carboplatin-based combination therapy with reduced dosage is considered the standard of care for NSCLC patients with PS 2 and without druggable genetic alterations. In the 2022 NCCN Guidelines Version 4 2022, the combination of carboplatin and pemetrexed for non-squamous NSCLC and the combination of carboplatin and albumin-bound paclitaxel, carboplatin and gemcitabine, or carboplatin and paclitaxel for squamous NSCLC are considered the preferred systemic therapy [19]. However, the evidence for cytotoxic chemotherapy for NSCLC with PS 2 is not sufficient. Efficacy is limited, and toxicity remains a major concern, especially in carboplatin-based combination therapy, with occasional treatment-related deaths.

### 2.2. Anti-PD-1/PD-L1 Antibody Monotherapy

ICIs are a promising treatment for patients with PS 2 because of lower incidence of severe toxicity compared to cytotoxic chemotherapy. However, data on the efficacy and safety of anti-Programmed cell Death-1 (PD-1)/PD-1 Ligand 1 (PD-L1) antibodies for NSCLC with PS 2 are much more limited than for cytotoxic chemotherapy, with only a few reports of interventional studies, let alone randomized trials. For advanced or recurrent NSCLC with high PD-L1 expression, anti-PD-1/PD-L1 antibody monotherapy is the standard of care as first-line therapy. However, in the pivotal KEYNOTE-024 and IMpower110 studies, only patients meeting PS 0-1 as eligibility criteria were enrolled [20,21], so the efficacy for NSCLC with PS 2 is currently uncertain. 

In recent years, results from several interventional trials have been reported (Table 2). The phase II Checkmate-171 trial of nivolumab as second-line therapy, which enrolled 811 squamous NSCLC overall, included 12.7% (103 patients) with ECOG PS 2 [22]. Patients with PS 2 had a median OS of 5.2 months, which was shorter than the overall and elderly population (10–11 months). The incidence of treatment-related adverse events of grade 3–4 was lower in patients with PS 2 (6.8%) compared to 13.9% overall, suggesting that the drug could be administered relatively safely. The CheckMate-153 study is a phase IIIB/IV study evaluating the safety and efficacy of nivolumab in patients with stage IIIB or IV NSCLC and an ECOG PS of 0 to 2 after disease progression at least one systemic therapy [23]. Among 1426 treated patients in this study, 128 (9%) had an ECOG PS of 2. Incidences of severe treatment-related adverse events in patients with PS 2 were comparable to the overall population and other subgroups. Patients with PS 2 had a median OS of 4 months, which tended to be shorter than the overall population (9.1 months) and patients aged 70 years or older (10.3 months). The global Phase III/IV TAIL study was designed to evaluate the safety and efficacy of atezolizumab monotherapy in a broad range of previously treated NSCLC patients, including those not included in the pivotal trials [24]. The study included 61 patients with ECOG PS 2. While the incidence of treatment-related serious adverse events in patients with PS 2 tended to be higher (14.8%) than in the overall population (7.8%), the incidence of immune-related adverse events (irAEs) was 8.2%, similar to that of the overall population (8.3%). The median PFS for patients with PS 2 was 1.7 months and median OS was 3.5 months. Moreover, the phase II PePS2 study of pembrolizumab monotherapy in 60 patients with NSCLC with ECOG PS 2 included 24 (40%) given as first-line therapy and 25% (15 patients) with PD-L1 TPS ≥ 50% [25]. In this study, pembrolizumab monotherapy resulted in toxicity requiring treatment delay or discontinuation in 28%. Treatment-related adverse events of grade ≥ 3 occurred in 15% of patients. No grade 5 treatment-related adverse events were observed. Recently, the results of the Phase III IPSOS trial comparing atezolizumab and single-agent chemotherapy (gemcitabine or vinorelbine) in patients with locally advanced or metastatic NSCLC who were ineligible for first-line platinum-based chemotherapy because of poor PS (≥2) or elderly patients with comorbidities were reported [26]. In this study, atezolizumab significantly improved OS compared to chemotherapy (HR, 0.78, 95%CI: 0.63–0.97, *p* = 0.028), with consistent benefits in key subgroups including PD-L1 expression levels, histology, and PS. With 76% (344/453) having ECOG PS 2, subgroup analysis of PS 2 showed a trend toward better OS in the atetzolizumab group, but no statistically significant difference (HR 0.86, 95% CI: 0.67–1.10). 

The results of these studies consistently suggested that anti-PD-1/PD-L1 antibody monotherapy can be relatively safe to administer, even to patients with PS 2. Regarding efficacy, results varied from study to study. While the PePS2 study of anti-PD-1 antibody pembrolizumab showed relatively favorable efficacy, the CheckMate-171 and CheckMate-153 studies of nivolumab, also an anti-PD-1 antibody, showed median OS was only 4.0–5.4 months. Even for anti-PD-L1 antibody atezolizumab, which showed favorable efficacy in the recent IPSOS trial, a previously reported TAIL study showed limited efficacy with a median PFS of 1.7 months and median OS of 3.5 months. Since poor PS is a poor prognostic factor for advanced NSCLC, the shorter PFS and OS of ICI for NSCLC with PS 2 in the aforementioned intervention trials may not be surprising. However, several reports suggested that the antitumor effect of anti-PD-1/PD-L1 antibody monotherapy itself is lower in patients with PS 2 than in those with PS 0-1. In a multicenter, retrospective study of NSCLC with PD-L1 ≥ 50% treated with pembrolizumab as first-line therapy at the Dana Farber Cancer Institute, Memorial Sloan Kettering Cancer Center, and University of Texas MD Anderson Cancer Center, patients with PS 2 had not only a significantly shorter OS (7.4 months vs. 20.3 months; HR 0.42, 95% CI 0.26 to 0.68; *p* < 0.001) but also a lower objective response rate (ORR) (25.6% vs. 43.1%; *p* = 0.04) and shorter PFS (4.0 months vs. 6.6 months; HR 0.70, 95% CI 0.47 to 1.06; *p* = 0.09) [27]. Therefore, there is a need to elucidate the causes of the low efficacy of ICI for PS 2 patients and to further improve treatment outcomes.

### 2.3. Combination Therapy with ICI and Cytotoxic Chemotherapy 

Currently, combination therapy with platinum-based chemotherapy and anti-PD-1/PD-L1 antibody is the standard of care for advanced NSCLC patients with PS 0-1 (especially PD-L1 < 50%). However, pivotal phase III trials of platinum-based chemotherapy and anti-PD-1/PD-L1 antibody combinations have all enrolled only patients meeting PS 0-1 as eligibility criteria [28,29,30]. Therefore, the efficacy and safety of Platinum-based chemotherapy + anti-PD-1/PD-L1 antibody as first-line therapy for advanced NSCLC with PS 2 is unclear. Even with platinum-based chemotherapy alone, there are concerns about toxicity, and the added risk of immune-related adverse events may be too great to ignore. To date, there is no clear rationale for recommending combination therapy with platinum-based chemotherapy and anti-PD-1/PD-L1 antibody for NSCLC patients with PS 2.

### 2.4. Combination Therapy with Nivolumab and Ipilimumab 

Combination therapy with nivolumab and ipilimumab, an anti-Cytotoxic T lymphocyte-associated antigen (CTLA)-4 antibody, has demonstrated long-term survival benefits across cancer types, beginning with its approval for the first-line treatment of malignant melanoma and renal cell carcinoma. In the randomized phase III CheckMate-227 trial part 1 for untreated advanced NSCLC, patients with PD-L1 ≥ 1% (part 1a, N = 1189) showed a significant OS benefit in the nivolumab plus ipilimumab group versus the chemotherapy group (hazard ratio: 0.79, 95% CI 0.67–0.93) [31]. Furthermore, patients with PD-L1 < 1% (Part 1b, N = 550) also showed significantly longer OS in the nivolumab plus ipilimumab group versus the chemotherapy group (hazard ratio: 0.64, 95% CI: 0.51–0.81), and the 5-year OS rate was 19% in the nivolumab plus ipilimumab group, better than in the chemotherapy + Nivolumab group (10%) and chemotherapy group (7%). The nivolumab plus ipilimumab combination group reported an increased frequency of cutaneous, endocrine, gastrointestinal, and hepatic irAEs compared to the nivolumab monotherapy group, but the irAEs were mostly known events and were reported to be manageable. Moreover, the nivolumab plus ipilimumab group showed improvement in both the average symptom burden index (ASBI), as assessed by the Lung Cancer Symptom Scale (LCSS), and scores for six specific symptoms, including cough, dyspnea, and fatigue, compared with the chemotherapy group [32]. With a long-term survival benefit and good tolerability, as well as favorable data on reported outcomes (PRO), nivolumab plus ipilimumab is promising for advanced NSCLC with poor PS.

The results of two intervention studies of nivolumab and ipilimumab for NSCLC with PS 2 have been reported (Table 2). CheckMate-817 trial was a multi-cohort, single-arm, phase IIIb study evaluating the safety and efficacy of nivolumab plus ipilimumab in advanced NSCLC, with cohort A enrolling 391 patients with ECOG PS 0-1 and cohort A1 including 139 patients with ECOG PS 2 and 59 patients with ECOG PS 0-1 but certain comorbidities (asymptomatic untreated brain metastases, hepatic or renal dysfunction, or HIV) [33]. In this study, there was no difference in the incidence and severity of irAEs in patients with PS 2 within cohort A1 (N = 139) compared to those with PS 0-1 in cohort A (N = 391). Furthermore, patients with PS 2 within cohort A1 had a median OS of 9.0 months (95% CI 5.5–12.9) and a 1-year OS rate of 44%. However, as expected, while inferior to the results for patients with PS 0-1 within cohort A (median OS of 17.0 months and 1-year OS rate of 60%, similar to the CheckMate-227 trial), these results show promise for the tolerability and efficacy of nivolumab plus ipilimumab in NSCLC with PS 2. 

In contrast, another trial showed skeptical results for nivolumab plus ipilimumab for NSCLC with ECOG PS 2. The Energy-GFPC 06-2015 study was a randomized phase III study of nivolumab and ipilimumab versus carboplatin-based doublet in first-line treatment of PS 2 or elderly (≥70 years) patients with advanced NSCLC [34]. Patients were stratified by age (≥70 vs. <70), PS (0/1 vs. 2) and histologic type (squamous vs. non-squamous) and were randomized 1:1. This trial was terminated after enrolling 204 patients because a preplanned interim analysis showed a risk of futility, especially in patients with PS 2 (HR 1.8, 95% CI, 0.99–3.3). For patients with PS 2 (N = 79), the median OS in the nivolumab plus ipilimumab and chemotherapy groups was 2.9 months (95% CI 1.4–4.8) vs. 6.1 months (3.5–10.4), respectively (*p* = 0.22). 

With regard to the Energy-GFPC 06-2015 study, no definitive conclusions can be drawn from these results alone, since no detailed information such as irAE was presented and only a small number of patients with PS 2 were included in the study, about 40 patients in each group. However, even though it is a subgroup analysis, the data are from a randomized phase III trial and should not be taken lightly. More detailed examination of why the large discrepancy from the favorable results in cohort A1 of the CheckMate-817 study occurred, including patient background, treatment status, and the frequency and severity of irAEs, is warranted in the future.

**Table 2 cancers-14-04861-t002:** Key prospective studies of ICI for NSCLC with PS 2.

	Phase	Histology	Line	Regimen	N	Median OS (Month)	1-Year OS (%)	Median PFS (Month)	ORR (%)	TRAE with Grade 3–4 (%)	Ref
CheckMate-171	II	Sq NSCLC	≥2	Nivolumab	98	5.4	27	-	11	6	[22]
CheckMate-153	III/IV	NSCLC	≥2	Nivolumab	123	4.0	24	-	-	12	[23]
TAIL	III/IV	NSCLC	≥2	Atezolizumab	61	3.5	22	1.7	3	15	[24]
PePS2	II	NSCLC	1, 2	Pembrolizumab	27 (TPS < 1%)	8.1	-	3.7	11	28	[25]
					15 (TPS 1–49%)	12.6	-	8.3	33		
					15 (TPS 50%-)	14.6	-	12.6	47		
IPSOS	III	NSCLC	1	Atezolizumab	228	(HR 0.86, 95%CI 0.67–1.10)	-	-	-	-	[26]
				Gemcitabine or Vinorelbine	116		-	-	-	-	
CheckMate-817	IIIb	NSCLC	1	Nivolumab + Ipilimumab	139	9.0	44	3.6	19	24	[33]
Energy-GFPC 06-2015	III	NSCLC	1	Nivolumab + Ipilimumab	40	2.9	-	-	-	-	[34]
				Chemotherapy	39	6.1	-	-	-	-	

Abbreviations: ICI, immune checkpoint inhibitor; NSCLC, non-small cell lung cancer; PS, performance status; OS, overall survival; PFS, progression free survival; ORR, objective response rate; TRAE, treatment-related adverse event; TPS, tumor proportion score; HR, hazard ratio; CI, confidence interval.

## 3. Challenges of ICI for NSCLC with PS 2

### 3.1. Why Is ICI Less Effective for Patients with PS 2?

According to the results of several interventional studies reported to date, anti-PD-1/PD-L1 antibody monotherapy and the combination of nivolumab plus ipilimumab have demonstrated a certain level of safety with no apparent increase in the frequency or severity of irAEs, even in NSCLC patients with PS 2. However, efficacy data such as OS and PFS were poor in most of the trials, suggesting that PS 2 patients are a population with low efficacy of ICI, even discounting their originally poor prognosis. Further investigation into the causes of low efficacy and improvement of outcomes are needed for ICI to become widely used as the standard of care for NSCLC with PS 2. 

The worsening of PS may be due to a variety of factors, including age-related changes, presence of comorbidities, and nutritional status, in addition to the disease activity of the cancer itself. Therefore, ECOG PS 2 is a highly heterogenous patient population, and it may be unreasonable to consider treatment strategies in a vacuum. A useful cut-off for predicting the efficacy of ICI considering heterogeneity in the PS 2 patient population has not yet been established. 

A recent report has shown that the efficacy of ICI varies greatly when patients with PS 2 are divided according to the cause of their PS decline. In a multicenter retrospective study (GOIRC-2018-01) of 1st line pembrolizumab in advanced NSCLC with PS 2 and PD-L1 ≥ 50%, patients with PS 2 due to comorbidities had significantly better outcomes than those with PS 2 due to the disease burden of NSCLC itself (6-month PFS rate 49% vs. 19%, median PFS 5.6 months vs. 1.8 months, OS 11.8 months vs. 2.8 months) [35]. These results are important data to determine the cause of the low efficacy of ICI in NSCLC with PS 2. Elucidating the mechanism of “disease burden” that diminishes the efficacy of ICIs is important not only for appropriate patient selection when administering ICIs for NSCLC with PS 2, but also for developing additional treatment strategies for patients who are less likely to respond to ICIs.

### 3.2. Impact of Cancer Cachexia on NSCLC with PS 2 

By clarifying the detailed mechanism of the “disease burden” that diminishes the efficacy of ICI, and by investigating ways to deal with it, we may be able to make better use of ICI in patients with advanced NSCLC at PS 2. The following three specific mechanisms of the “disease burden” may be considered (Figure 1).

First, it has been suggested that overproduction of inflammatory cytokines such as tissue growth factor (TGF)-β and interleukin (IL)-6 due to cross-reactivity between cancer and host may attenuate the efficacy of ICIs. By inhibiting T cell infiltration, TGF-β diminishes the anti-tumor effects through blockade of PD-1/PD-L1 signaling [36]. It has also been reported that the blockade of IL-6 exerts a synergistic antitumor effect when blocked simultaneously with PD-1/PD-L1 signaling [37]. In addition, chronic inflammation can cause immune escape of tumor cells through mechanisms such as T cell exhaustion [38]. 

Second, a decrease in myokine and PGC-1α due to reduced skeletal muscle mass may attenuate the efficacy of ICI. Several reports suggested that myokine, produced and secreted by skeletal muscle, and PGC-1α, a transcriptional cofactor involved in regulating mitochondrial neogenesis and slow-twitch muscle fiber formation, affect the anti-tumor immune response [39,40,41,42]. 

Third, weight loss due to tumor disease burden may result in less PD-L1-positive CD8-positive T cells, making ICI less effective. In vivo, obese mice had a significantly higher percentage of PD-L1-positive CD8-positive T cells in their tumors. In addition, PD-1 inhibitor treatment resulted in greater tumor shrinkage in obese mice compared to control mice [43]. Although the mechanism is not fully elucidated, it is possible that reduced nutritional status may cause decreased methionine uptake, triggering reduced T cell activity [44]. Malnutrition has also been suggested to worsen treatment delivery of carboplatin-based combination therapy for NSCLC with PS 2 and adversely affect antitumor efficacy [45]. The same may be true for ICI, and further validation is warranted.

These mechanisms of (1) overproduction of various cytokines derived from cancer and the associated inflammatory response, (2) loss of skeletal muscle mass and weight loss, and (3) impaired nutritional status might indeed indicate the development of “cancer cachexia”. In a non-interventional, cross-sectional study led by Gustave Roussy, as many as 67.6% of patients with advanced NSCLC with PS 2, including previously treated cases, had cancer cachexia [46]. Although only speculative at this point, the development of cancer cachexia might be a major cause of the poor efficacy of ICI in advanced NSCLC with PS 2. 

Actually, there have been several reports suggesting an association between cancer cachexia and low efficacy of ICI. In a retrospective study of advanced NSCLC with PS 0-1 treated with anti-PD-1/PD-L1 antibody monotherapy at the Shizuoka Cancer Center, patients who met the definition of cancer cachexia (≥5% weight loss within 6 months) prior to treatment initiation had significantly lower ORR (15% vs. 57%, *p* < 0.001) and significantly shorter PFS (2.3 months vs. 12.0 months, *p* < 0.001) [47]. In this study, PFS was shorter in patients meeting the definition of cancer cachexia, even with high PD-L1 expression, and was not significantly different from that with low PD-L1 expression. 

Therefore, it may be useful to focus on the presence or absence of cancer cachexia when considering treatment strategies for advanced NSCLC in PS 2. For patients without cancer cachexia, ICI is expected to be effective and may be aggressively considered, taking into account the degree of PD-L1 expression. On the other hand, for patients with cancer cachexia, expectations for the efficacy of ICI may be low. However, cancer cachexia has also been reported to be associated with high toxicity and low tolerance of chemotherapy [48]. Therefore, for NSCLC with PS 2 and cancer cachexia, the establishment of special treatment strategies to enhance the efficacy of ICI is warranted.

### 3.3. How to Make ICI More Effective for NSCLC with PS 2?

An anti-CTLA-4 antibody, ipilimumab, may overcome the decrease in T-cell activity and CD8-positive T cells due to decreased methionine uptake caused by impaired nutritional status, which may contribute to the low efficacy of anti-PD-1/PD-L1 antibody monotherapy for NSCLC in PS 2 (Figure 1). Ipilimumab activates CD8-positive T cells by releasing the CTLA-4-mediated brake during the priming phase. Preclinical data show that the percentage of CD8-positive T cells in tumor-infiltrating lymphocytes increases with ipilimumab compared to anti-PD-1 antibody monotherapy [49]. Ipilimumab can also increase CD4-positive T cells, and the associated increase in memory T cells may produce long-term effects [50]. In addition, a large proportion of PS 2 NSCLC are elderly. Preclinical data report an increase in regulatory T cells in the peritumoral environment with aging [51]. As CTLA-4 is expressed on regulatory T cells, suppression of regulatory T cells and reduction in regulatory T cells in tumor tissue via antibody-dependent cell-mediated cytotoxicity by ipilimumab may also enhance an antitumor effect [52]. Combination therapy with nivolumab plus ipilimumab for this patient population remains controversial, as there was a large discrepancy between the efficacy results of the Check-Mate-817 trial and the Energy-GFPC 06-2015 study. Accumulation of data in more patients is warranted in the future. At the very least, the concomitant use of the anti-CTLA-4 antibody ipilimumab clearly increases the incidence of irAEs. Therefore, more careful risk–benefit assessment and strict management of irAEs are required.

Treatment for cancer cachexia, as suggested in the previous section, may also enhance the efficacy of ICI for NSCLC with PS 2 (Figure 1). Anamorelin, a ghrelin receptor agonist, has been shown to increase body weight and lean body mass and improve appetite in NSCLC patients with cancer cachexia in a Japanese phase II study (ONO-7643-04) [53]. Based on these results, the drug was approved for the treatment of cancer cachexia in Japan on 22 January 2021, ahead of any other country in the world. Ghrelin also has anti-inflammatory effects by suppressing inflammatory cytokines induced via NF-κB [54]. In NSCLC patients with PS 2 and cancer cachexia, ICI may be inherently more effective when combined with anamorelin, which is expected to increase body weight and stimulate appetite, as well as have anti-inflammatory effects. Moreover, ghrelin has been reported to promote lymphocyte maturation and differentiation in primary lymphoid organs (bone marrow and thymus) and to inhibit age-related regression of the thymus [55]. This may also help to enhance the efficacy of ICI. Currently, however, there are no data on the combination of anamorelin and ICI, and safety and other factors need to be carefully verified.

Recently, a novel research hypothesis has been proposed that exercise sensitizes the anti-tumor effect of ICI (Figure 1). The study using a pancreatic cancer model suggested that aerobic exercise restricts pancreatic tumor growth by enhancing anti-tumor immunity, driven by IL-15Rα+ CD8 T cells [56]. Furthermore, a study in a mouse model of breast cancer suggested that exercise training normalizes tumor vasculature and enhances the ICI effect by increasing CD8+ T cell infiltration via CXCR3 signaling [57]. Discussion of the heterogeneity of the PS 2 population from the aspect of physical activity is also very important for considering the efficacy of ICI.

## 4. Conclusions and Future Perspectives

In this review, we summarized the current evidence for drug therapy for NSCLC patients with PS 2 and without druggable genetic alterations, followed by a discussion of treatment strategies that take into account the heterogeneity of PS 2. It was suggested that the efficacy of ICI for NSCLC with PS 2 may be estimated by focusing on the presence or absence of cancer cachexia, but on the other hand, how to make ICI work for patients with cancer cachexia is a future challenge. Accumulation of data from a larger number of patients is required.

In order to establish optimal treatment strategies for advanced NSCLC with PS 2 utilizing ICI, further interventional trials should be designed and conducted in a larger number of patients. Future planned trials would need to take PS 2 heterogeneity into account in their study design or pre-specify subgroup analyses using various frailty measures. 

## Figures and Tables

**Figure 1 cancers-14-04861-f001:**
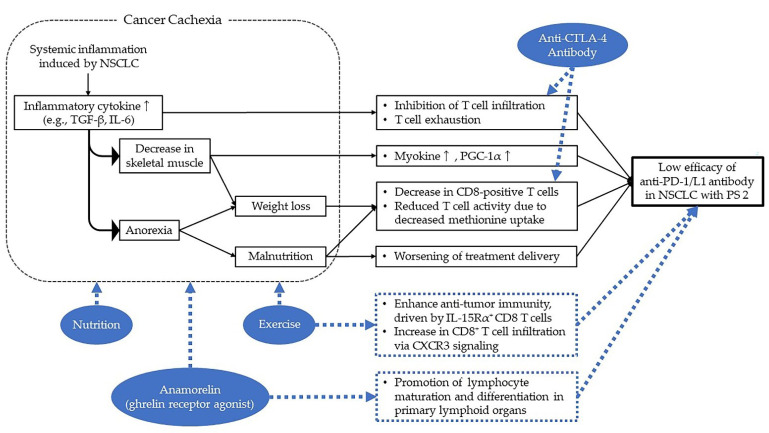
Possible causes and potential treatments for low efficacy of ICI in NSCLC with PS 2. Abbreviations: NSCLC, non-small cell lung cancer; ICI, immune checkpoint inhibitor; PS, performance status; PD-1, Programmed cell Death-1; PD-L1, Programmed cell Death-1 Ligand 1; CTLA, Cytotoxic T lymphocyte-associated antigen.

**Table 1 cancers-14-04861-t001:** Key randomized trials of cytotoxic chemotherapy for NSCLC with PS 2.

Heading	Phase	Regimen	N	Median OS (Month)	1-Year OS (%)	Median PFS (Month)	ORR (%)	Ref
Lienbaum et al. (CALGB 9730)	III	PTX	50	2.4	10	-	10	[12]
		CBDCA + PTX	49	4.7	18	-	24	
Kosmidis et al.	II	GEM	47	4.8	18	3	4	[13]
		CBDCA + GEM	43	6.7	20	4.1	14	
Langer et al. (ECOG 1599)	II	CBDCA + PTX	49	6.2	19	3.5	14	[14]
		CDDP + GEM	54	6.9	25	3	23	
Reynolds et al.	III	GEM	85	5.1	21	2.7	16	[15]
		CBDCA + GEM	85	6.7	31	3.8	43	
Saito et al. (WJTOG0004)	II	GEM + VNR	43	6.0	28	2.7	21	[16]
		CBDCA + PTX	41	5.9	22	2.9	29	
Morabito et al. (CAPPA-2)	III	GEM	28	3.0	NR	1.7	4	[17]
		CDDP + GEM	28	5.9	NR	3.3	18	
Zukin et al.	III	PEM	102	5.3	22	2.8	11	[18]
		CBDCA + PEM	103	9.3	40	5.8	24	

Abbreviations: PTX, paclitaxel; CBDCA, carboplatin; GEM, gemcitabine; VNR, vinorelbine; PEM, pemetrexed.

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
