# Peer review of "Pharmacotherapy for Advanced Non-Small Cell Lung Cancer with Performance Status 2 without Druggable Gene Alterations: Could Immune Checkpoint Inhibitors Be a Game Changer?"

_cancers, 2022, doi:10.3390/cancers14194861_

Round 1

Reviewer 1 Report

The authors present an extensive and deep review focused on this relevant topic. Their approach covers both clinical aspects and underlying molecular mechanisms. They adress challenges and propose strategies in a conservative and rigurous manner. It is well written and structured, I really enjoyed reading it. It is, indeed, a nice paper.

My only suggestion is regarding the description of the IPSOS trial. Even though the differences in the PS2 subgroup analysis did not reach statistical significance, the trial was positive and was not powered for this analysis. I think this aspect should be pointed out to avoid confusion.

Congratulations for the good work.

Author Response

Thank you for your kind comments and advice about our manuscript.

Based on the suggestions received from you, we have made the following changes to the manuscript.

<General comment>

The authors present an extensive and deep review focused on this relevant topic. Their approach covers both clinical aspects and underlying molecular mechanisms. They adress challenges and propose strategies in a conservative and rigurous manner. It is well written and structured, I really enjoyed reading it. It is, indeed, a nice paper.

(Response)

Thank you very much for your encouraging comments.

To make it easier for readers to understand the overview of this paper, we have created and added “Figure 1” summarizing possible causes and potential treatments for low efficacy of ICI in NSCLC with PS 2

(Changes)

[p.7] By clarifying the detailed mechanism of the "disease burden" that diminishes the efficacy of ICI, and by investigating ways to deal with it, we may be able to make better use of ICI in patients with advanced NSCLC at PS 2. The following three specific mechanisms of the "disease burden" may be considered (Figure 1).

[p.8] An anti-CTLA-4 antibody, ipilimumab, may overcome the decrease in T-cell activity and CD8-positive T cells due to decreased methionine uptake caused by impaired nutritional status, which may contribute to the low efficacy of anti-PD-1 / PD-L1 antibody monotherapy for NSCLC in PS 2 (Figure 1).

[p.8] Treatment for cancer cachexia, as suggested in the previous section, may also enhance the efficacy of ICI for NSCLC with PS 2 (Figure 1).

[p.9] Figure 1. Possible Causes and Potential Treatments for Low efficacy of ICI in NSCLC with PS 2.

<Comment 1>

My only suggestion is regarding the description of the IPSOS trial. Even though the differences in the PS2 subgroup analysis did not reach statistical significance, the trial was positive and was not powered for this analysis. I think this aspect should be pointed out to avoid confusion.

(Response)

Thank you very much for your important comments. Following your advice, I have revised the text in the relevant section as follows.

(Changes)

[p.4] Recently, the results of the Phase III IPSOS trial comparing atezolizumab and single-agent chemotherapy (gemcitabine or vinorelbine) in patients with locally advanced or metastatic NSCLC who were ineligible for first-line platinum-based chemotherapy because of poor PS (≥2) or elderly patients with comorbidities were reported [26]. In this study, atezolizumab significantly improved OS compared to chemotherapy (HR, 0.78, 95%CI: 0.63-0.97, P=0.028), with consistent benefits in key subgroups including PD-L1 expression levels, histology, and PS. With 76% (344 / 453) having ECOG PS 2, subgroup analysis of PS 2 showed a trend toward better OS in the atetzolizumab group, but no statistically significant difference (HR 0.86, 95%CI: 0.67-1.10).

Reviewer 2 Report

The manuscript entitled:"Pharmacotherapy for advanced Non-Small Cell Lung Cancer 2 with performance status 2 without druggable gene alterations: 3 could immune checkpoint inhibitors be a game changer?" focused on a systemic revision of literature data about the ICi in the first line PS 2 NSCLC patients is a tiley relevant manuscrit aimed to elucidate the best therapuetical approach in a specific clinical setting. In my opinion, some minor considerations should be implemented to accept this paper for the publication

- In the methodological section, pelase, could the authors list the drivers oncogene used to filter NSCLC patients from clinical analysis? In my opinion, this point may improve thereadibility of the manuscript

- In the text, could the authors show if different ICIs drugs may differentially impact on the clinical response for NSCLC patients? Could this aspect may improve the drug selection for NSCLC patients?

- Could the authors analyze and compare adversae events derived from these two clinical approaches?

Author Response

Thank you for your kind comments and advice about our manuscript.

Based on the suggestions received from you, we have made the following changes to the manuscript.

<Comment 1>

- In the methodological section, please, could the authors list the driver oncogene used to filter NSCLC patients from clinical analysis? In my opinion, this point may improve the readability of the manuscript

(Response)

Thank you very much for your important comments. Strictly speaking, most of the trials discussed in this review article do not require measurement of rare druggable genetic alterations (e.g., MET, RET, NTRK, etc.) other than the relatively frequent ones such as EGFR and ALK. However, all trials exclude patients who have been previously treated with targeted therapies. We have added the following at the end of the Introduction section.

(Changes)

[p.2] As a reminder, strictly speaking, most of the trials discussed in this review article do not require measurement of rare druggable genetic alterations (e.g., MET, RET, NTRK, etc.) other than the relatively frequent ones such as EGFR and ALK. However, all trials exclude patients who have been previously treated with targeted therapies.

<Comment 2>

- In the text, could the authors show if different ICIs drugs may differentially impact on the clinical response for NSCLC patients? Could this aspect may improve the drug selection for NSCLC patients?

(Response)

Thank you very much for your important comments. As for anti-PD-1 and anti-PD-L1 antibodies, it is difficult to realistically differentiate their use with respect to their efficacy against the NSCLC with PS 2 population. While the PePS2 study of anti-PD-1 antibody pembrolizumab showed relatively favorable efficacy, the CheckMate-171 and CheckMate-153 studies of nivolumab, also an anti-PD-1 antibody, showed median OS was only 4.0-5.4 months. Even for anti-PD-L1 antibody atezolizumab, which showed favorable efficacy in the recent IPSOS trial, previously reported TAIL study showed limited efficacy with a median PFS of 1.7 months and median OS of 3.5 months. We have added the following text in section "2.2 Anti-PD-1 / PD-L1 antibody monotherapy".

(Changes)

[p.4] Regarding efficacy, results varied from study to study. While the PePS2 study of anti-PD-1 antibody pembrolizumab showed relatively favorable efficacy, the CheckMate-171 and CheckMate-153 studies of nivolumab, also an anti-PD-1 antibody, showed median OS was only 4.0-5.4 months. Even for anti-PD-L1 antibody atezolizumab, which showed favorable efficacy in the recent IPSOS trial, previously reported TAIL study showed limited efficacy with a median PFS of 1.7 months and median OS of 3.5 months.

<Comment 3>

- Could the authors analyze and compare adverse events derived from these two clinical approaches?

(Response)

Thank you very much for your comments. As for anti-PD-1 and anti-PD-L1 antibodies, there is no apparent difference in terms of efficacy or safety. However, the incidence of irAEs is clearly increased when ipilimumab, an anti-CTLA-4 antibody, is used in combination with nivolumab. The combination of nivolumab plus ipilimumab is a promising treatment option for NSCLC with PS 2 compared to anti-PD-1 / PD-L1 antibody monotherapy, but requires more careful risk-benefit assessment and strict irAE management. We have added the following text in section "3.3. How to make ICI more effective for NSCLC with PS 2?".

(Changes)

[p8-9] At the very least, the concomitant use of the anti-CTLA-4 antibody ipilimumab clearly increases the incidence of irAEs. Therefore, more careful risk-benefit assessment and strict management of irAEs are required.

Reviewer 3 Report

This is a well written review that summarizes the current data on the treatment of frail patients with NSCLC that present a huge challenge in our daily practice. Recent studies are thorougly analyzed with regard to their implication for this patient population and the authors also undertake efforts to explain the underlying mechanisms of the reduced efficacy of ICI in patients with poor performance status.